# Housekeeping in the Hydrosphere: Microbial Cooking, Cleaning, and Control under Stress

**DOI:** 10.3390/life11020152

**Published:** 2021-02-17

**Authors:** Bopaiah Biddanda, Deborah Dila, Anthony Weinke, Jasmine Mancuso, Manuel Villar-Argaiz, Juan Manuel Medina-Sánchez, Juan Manuel González-Olalla, Presentación Carrillo

**Affiliations:** 1Annis Water Resources Institute, Grand Valley State University, Muskegon, MI 49441, USA; weinkea@mail.gvsu.edu (A.W.); mancusja@mail.gvsu.edu (J.M.); 2School of Freshwater Sciences, University of Wisconsin-Milwaukee, Milwaukee, WI 53204, USA; dila@uwm.edu; 3Departamento de Ecología, Facultad de Ciencias, Universidad de Granada, 18071 Granada, Spain; mvillar@ugr.es (M.V.-A.); jmmedina@ugr.es (J.M.M.-S.); 4Instituto Universitario de Investigación del Agua, Universidad de Granada, 18071 Granada, Spain; jmolalla@ugr.es (J.M.G.-O.); pcl@ugr.es (P.C.)

**Keywords:** ecology, ecosystem structure and function, aquatic microbes, stressor interactions, perturbations, microbiome, biogeochemistry

## Abstract

Who’s cooking, who’s cleaning, and who’s got the remote control within the waters blanketing Earth? Anatomically tiny, numerically dominant microbes are the crucial “homemakers” of the watery household. Phytoplankton’s culinary abilities enable them to create food by absorbing sunlight to fix carbon and release oxygen, making microbial autotrophs top-chefs in the aquatic kitchen. However, they are not the only bioengineers that balance this complex household. Ubiquitous heterotrophic microbes including prokaryotic bacteria and archaea (both “bacteria” henceforth), eukaryotic protists, and viruses, recycle organic matter and make inorganic nutrients available to primary producers. Grazing protists compete with viruses for bacterial biomass, whereas mixotrophic protists produce new organic matter as well as consume microbial biomass. When viruses press remote-control buttons, by modifying host genomes or lysing them, the outcome can reverberate throughout the microbial community and beyond. Despite recognition of the vital role of microbes in biosphere housekeeping, impacts of anthropogenic stressors and climate change on their biodiversity, evolution, and ecological function remain poorly understood. How trillions of the smallest organisms in Earth’s largest ecosystem respond will be hugely consequential. By making the study of ecology personal, the “housekeeping” perspective can provide better insights into changing ecosystem structure and function at all scales.

## 1. Foreword: New Approach to an Old Paradigm

“Microbes, the unseen strands in the food web, are major producers as well as consumers of matter and energy in the sea”—Lawrence Pomeroy (1974) [1].

Years before the tiniest microbes, such as bacteria and viruses, could be directly observed in natural waters, an insightful inference was made regarding the key role that microbes play in the sea. Based on unprecedented size-fractionated photosynthesis and respiration measurements made in the surface waters and the deep ocean during the 1960s and 1970s that revealed the bulk of the activity occurred in the invisible microbial fraction, Lawrence Pomeroy posited that microbes are both major producers as well as consumers of matter and energy in the sea [1]. In that BioScience paper titled “The Ocean’s Food Web, a Changing Paradigm”, Pomeroy elaborated his thesis in plain and easy to relate language that a layperson could grasp. Subsequently, his visionary ideas have launched numerous studies—many ongoing—that have resulted in major revisions of our understanding of how aquatic food webs are structured and ecosystems function [2]. Whereas many of these ensuing developments in aquatic microbial ecology are fairly well-known, herein, we discuss them under a new, simple, and relatable analogy (house, household, housekeeping, and economy) wherein the “unseen strands” play out in the underwater world. The housekeeping analogy should be useful in the study of all kinds of ecosystems—spanning microbial to planetary scales.

In a classic paper titled, “How to write consistently boring scientific literature”, author Kaj Sand-Jensen begins with an account of how the first draft of a student’s Ph.D. thesis that was written in “a personal style, slightly verbose but with humoristic tone and thoughtful side-tracks” was transformed by departmental guidelines into a successful but “technical, boring and impersonal scientific writing […] turning a gifted writer into a dull scientist” [3]. In trying to resolve the problem, “Why are scientific publications so boring?”, Sand-Jensen advocates 10 steps—such as including originality and personality, explanatory illustrations, reasoned speculation, relatable analogies and humor, etc.—to convey the genuine excitement of scientific findings to the readership. Fortunately, many noteworthy efforts are already underway, bringing compelling science stories directly into our laboratories [4,5,6], classrooms [7,8,9], and living rooms [10,11,12]. The authors of this review are of the opinion that better science communication will not only advance science, but also help better connect science with society—a connection sorely deficient in today’s divided world where many find scientists dry and distant, and the science itself, unrelatable. Our light-hearted but rigorous review is a step in this direction.

## 2. Ecology: Study of the House

Ecology is the study of the house—the prefix “eco” being derived from the common etymological Greek word “Oikos”, meaning “home, house, household, or economy”. The term for the discipline of “ecology” or “oekologie” was coined first by the German naturalist Ernst Haeckel to describe the relation of organisms with their environment [13]. Haeckel linked the two words “ecology” and “economy” both derived from “oikos”, to describe one common discipline involving both the study of the household and the management of the household. Haeckel’s idea of ecology as the study of the household and its economy or management would inspire all those who subsequently studied nature—including human-dominated ecosystems. In more recent times, ecology has come to be defined as the study of interrelationships of organisms with each other and their environment, the economy of nature, or the biology of ecosystems. The American ecologist, Eugene Odum, described ecology as the “study of the structure and function of nature” [14].

However, today, we rarely come across ecological studies that view ecosystems, whether they are ponds or forests, oceans or the Earth, from a “house”, “household”, or “economy of housekeeping” perspective. Clearly, the “house, household, and housekeeping” perspective is key to attaining a more personal and fuller understanding of the world around us—the ecosystems we live in—from our homes, fields, lakes and forests, all the way to the planet itself. Herein, we attempt to make the case that “housekeeping” is a useful working concept for better science, education, and communication of ecosystem studies. In the following narrative, we employ a personal tone, a touch of humor, simple wit, and descriptive analogies to convey the housekeeping framework of aquatic ecosystems in particular and all ecosystems in general—a perspective that may become increasingly vital in an anthropogenically perturbed and rapidly changing world.

## 3. Remodeling the Big House

Over 3.5 billion years ago, microorganisms began keeping house here on Earth [15]. Today, an invisible network of sea and soil microbes blanket our biosphere. Ubiquitous and abundant, microorganisms have always been the most abundant life form on Earth [16], and this is why their daily activities have such a high impact on all ecosystems [15]. Microorganisms weave carbon, hydrogen, oxygen, nitrogen, phosphorus, sulfur, and other bioactive elements into the varied and functional metabolic pathways that operate from organism to ecosystem levels in the biosphere. They also provide humans with numerous ecosystem services essential to maintaining a healthy environment ranging from regulating the composition of Earth’s atmosphere to waste management [17,18]. In light of increasing anthropogenic stressors and climate change impacts, such as rapidly rising temperatures in lakes and oceans, what happens in the world’s waters will be consequential for the biosphere’s feedback on future climate [19,20,21].

The hydrosphere is the largest component of our biosphere. A focus on aquatic microbes is especially relevant considering they account for >90% of the aquatic biomass and are responsible for >95% of aquatic metabolism, thus playing a major role in the biospheric carbon cycle [18,20]. Although terrestrial and aquatic components of the biosphere process roughly equal amounts of carbon, recent analysis has revealed that freshwater ecosystems (which cover only about 3% of the planet’s surface) are globally significant hot spots for carbon cycling [22,23]. It is also becoming increasingly clear that freshwater bodies and their biota are globally-distributed sensitive sentinels of change, serving as an early warning system of climate and societal changes and harbingers of changes that are likely to be subsequently experienced by their larger and deeper marine counterparts. The following sections address the key housekeeping roles—cooking, cleaning, and control—that microbes perform in a rapidly changing and ever-dynamic water world (Figure 1).

## 4. Sunny Side Up: Fusion-Style Cooking with Supplements and Mixotrophy

To better relate to how aquatic microbes shape our modern household ecosystems, let us start with "cooking". Microorganisms are important players in moving nutrients and energy through food webs and biogeochemical cycles [24]. In a world primarily covered by water, it is not surprising that phytoplankton are responsible for roughly half of net photosynthesis globally [25,26]. They cook up sugary organic carbon dishes from carbon dioxide (CO_2_) and water to make biomass, and throw the oxygen away. The products of these workhorses are not just good eats for grazers but good breathing for rest of us metazoans, as about half of oxygen in every breath we take is courtesy of phytoplankton. Phytoplankton may be the smallest photosynthetic organisms on Earth, but their contribution to primary production in diverse ways makes them master chefs in the aquatic kitchen [27].

However, there is more than one way to excel in the aquatic kitchen. For instance, some cyanobacteria not only fix CO_2_ into foodstuffs for grazers, releasing oxygen in the process—they also spice up the whole affair. By enzymatically fixing atmospheric dinitrogen (N_2_) into a form that is bioavailable (ammonium), filamentous freshwater *Dolichospermum* (formerly *Anabaena*), marine *Trichodesmium*, and other unicellular planktonic cyanobacteria provide an essential nutrient whose absence would otherwise bring primary production to a crawl [28,29,30] (Figure 2).

The opposite of nitrogen fixation is the removal of reactive nitrogen, from the biosphere. This removal is conjured by microbial processes, mainly denitrification, which return N_2_ to the atmosphere along with small quantities of NO and N_2_O, two potent greenhouse gases that are involved in global warming, formation of acid rain, and destruction of the ozone layer [31]. Those processes of reactive nitrogen removal are performed mainly by anaerobic bacteria inhabiting environments with low O_2_ concentration, such as subsurface soils and sediments and anaerobic hypolimnion of lakes. Unexpectedly, active and diverse communities of denitrifying bacteria have been found in some well-oxygenated ecosystems, such as high-mountain lakes, where denitrification can explain the seasonal decline in nitrate concentration observed in the water column [32]. On the whole, aquatic denitrification may be on the rise coincident with increasing ocean deoxygenation, a trend that may greatly reduce productivity in nitrogen-limited waters by disadvantaging nitrogen-needy non-nitrogen fixing phytoplankton and heterotrophic bacteria—with impacts extending all the way up in the food chain [33].

Moreover, mixotrophy—ability to photosynthesize as well as graze—is widespread among planktonic prokaryotes as well as eukaryotes, and can confer significant advantages for acquiring food and nutrients in natural waters [34,35,36]. By combining autotrophic and heterotrophic lifestyles, mixotrophs avoid the light/nutrient limitations of phototrophs and the food/prey limitations of phagotrophs (Figure 3). This way of cooking and eating food generates a paradoxical relationship between the mixotrophs and bacteria, particularly relevant for those inhabiting oligotrophic clear-water ecosystems (e.g., high-mountain lakes, oceanic waters), which has been defined as “neither with nor without you” [37]. In this relationship, bacteria feed on dissolved organic carbon (e.g., sugars, organic acids) released by the mixotrophs (“without you, I cannot live”) but, simultaneously, are eaten by their own feeders (“with you, I die”). This relationship between mixotrophs and bacteria is comparable to other well-known mutualisms in nature, such as that between the Attine ants and fungi in rainforests. In natural waters, bacteria use organic matter released by themselves and their grazers, and are, in turn, eaten by them.

With this fusion style cooking, mixotrophs have an adaptive advantage compared to other microeukaryotes, such as strict autotrophs and heterotrophs. Thus, mixotrophs are flexible enough to adapt to their environment based on a feeding strategy that is most energetically profitable. By feeding on bacteria as a source of carbon and mineral nutrients, mixotrophs can overcome the stress caused by high UVR and warming that impairs the processes of photosynthesis and mineral nutrient uptake [37,38,39]. By combining phagotrophy with photosynthesis, mixotrophs can thrive with lower bacterial prey densities as well as under low light and nutrient availability than can the heterotrophic flagellates due to their enhanced functional redundancy.

Both food preparation and organismal growth at the primary producer level involve the manipulation of the chemical elements that constitute protein, carbohydrate, and fat. It is then time for the “gastronomic feast” by the primary consumers, who grow using the energy obtained from these “prepared” primary foods. This is how energy and nutrients transfer through the classic food web from phytoplankton to zooplankton to fish [40]. Consumption of smaller cells (bacteria, protozoa, phytoplankton) is necessary for production to flow from the microbial to macroscopic food web. We now know that copepods, cladocerans and rotifers, the dominant metazoan consumers in most natural waters, are both herbivores that feed on autotrophic phytoplankton and phagotrophs that ingest bacteria—particularly during their naupliar development or when phosphorus and phytoplankton biomass are limited [41]. From the smallest protozoans to the large metazoans, these intermediate trophic levels play a key role in the transfer of energy and nutrients to higher aquatic consumers including fish and whales [42].

Microbes in some extreme ecosystems practice other fusion cooking styles: e.g., cyanobacteria in the sulfidic sinkholes of the Laurentian Great Lakes and subglacial Antarctic lakes indulge in both oxygenic and anoxygenic photosynthesis [43,44], whereas the archaea inhabiting subseafloor sediments of North Atlantic Ocean couple mixotrophy and chemolithotrophy [45]. Thus, the housekeeping perspective will be useful for envisioning and exploring life in Earth’s many and varied extant ecosystems such as tiny cryoconites, ephemeral pitcher plants and the long-lived deep biosphere [46,47,48] and can even contribute testable household concepts for the exploration of life in extraterrestrial ecosystems [49].

## 5. When the Cook Is Out: TV Dinners and Raiding the Pantry

Sometimes, the cook is out, and an organism has to fend for itself—that is, primary producers are just not producing enough to support local demand. Some bacterioplankton bypass this via a light-activated protein, proteorhodopsin [50]. Proteorhodopsin grants the ability to utilize light for auxiliary ATP production, thus cushioning energy requirements when organic carbon substrates are in short supply. Due to the wide distribution and abundance of these photoheterotrophs in Earth’s waters, light-harvesting pigments might be seen as solar-powered TV dinners that just need a quick zap of light to provide bacteria and archaea the stamina necessary for the many housekeeping tasks at hand.

But when chefs are not cooking, it does not necessarily mean the cupboards are bare. A short burst of primary production when times are good can be too much for heterotrophs to consume, and some leftovers sink to deeper waters. Later, when the water-column mixes, the “ghost of production past” becomes available to heterotrophs residing in the surface waters. This is handy if primary production happened to be at a low point prior to mixing. The underwater pantry can also be stocked with organic matter imported from terrestrial sources or excess production from an earlier season [51]. Riverine and sediment contributions also include a dose of essential dietary supplements—nitrogen and phosphorus. So, when primary production gets up to speed, necessary nutrients are available.

## 6. Cleaning: Waste Busting and Recycling

In most households, cleaning jobs are hard to fill. Enter heterotrophic bacterioplankton—petite recyclers of the aquatic microbial household. Respiration by heterotrophic bacterioplankton dominates carbon flux in open waters, where their preferred meal, dissolved organic matter, contains the bulk of available organic carbon [51]. Only heterotrophic bacteria can efficiently utilize dissolved organic carbon as an entrée. If it was not for the protozoa who graze on bacterioplankton, moving secondary production to higher trophic levels, or viruses lysing their bacterial hosts, much of our watery planet’s carbon and essential nutrients would slowly be sequestered away by heterotrophic prokaryotes (Figure 4).

However, the utilization of dissolved organic carbon by microbes also depends on the main resource that limits microbial growth. Whether the limiting resource is organic carbon, mineral nutrients (such as phosphorus or nitrogen), or both, these have biogeochemical implications concerning how and how much organic carbon is recycled within the water column [52]. Recent research has examined the topic of co-limitation and the high complexity of co-limitation types. Results suggest that ecological interactions (e.g., competition, grazing or predation pressure) are key determinants of the nature of resource co-limitation that microbes experience in aquatic ecosystems [53].

As well as direct grazing of planktonic bacteria by protozoans, many bacteria feed into higher trophic levels by living in or on phytoplankton and detrital aggregates, commonly referred to as marine snow or lake snow (underwater snow, henceforth; Figure 5). Particles and detritus aggregate to form underwater snow, a food source for some fish, thus directly recycling protein-rich bacterial biomass to higher trophic levels. It is kind of a dormitory living situation—a few different lifestyles thrown together in some sticky carbohydrate-rich stuff—that creates hot spots of intense microbial and chemical activity in the water column [54]. A constant cycle of microbe-mediated detritus aggregation, decomposition and fragmentation in the water column plays a pivotal role in regulating the biological carbon pump in aquatic ecosystems [55,56]. Organic molecules are converted back to inorganic molecules as they are processed through the microbial loop (microbial trophic pathway wherein dissolved organic matter is returned to higher trophic levels by its incorporation into bacterial biomass), making these nutrients bioavailable for primary producers [24,57]. From the surface to the sediment, bacteria and protozoa tackle cleaning a “dirty dorm” with the enthusiasm of ardent recyclers.

## 7. Remote Control: Who Has Got It?

So, who’s got the remote control? In most households, that is a loaded question. In the aquatic microbial household, the answer lies in its smallest members—viruses. They move genes around within and between species, where “remote control” refers to horizontal gene transfer with genes functioning like “TV-Channels” that are being switched or modified [58,59]. Microbial genomes are subject to viral redesign in a process known as transduction. On occasion, when transfer includes an entire gene—voila, the host may acquire some new skills such as symbionts, and their communities may thrive [60,61]. Over time, such small transfers of DNA can add up to shape evolution and mediate the spread of genetically modified characteristics among organisms. For example, genes from widely distant organisms can occur in some genomes—the result of viral transactions over evolutionary time. Transduction is not the only method of horizontal gene transfer, but it may be a very important one in the aquatic world where viruses generally outnumber their hosts. The upside of viral gene transfer into host bacterial genomes (lysogeny), is enhanced fitness and survival of bacterial populations under changing environmental conditions. The downside is the often-collateral damage of your family being lysed (lytic)—a hefty price to pay for your gene movers.

In addition to serving as vectors for gene transfer among organisms, each day, viral activity kills about 20% of all aquatic microbes and about half of all aquatic bacteria, controlling plankton populations through both top-down and bottom-up mechanisms [9]. Freeing nutrients, shunting organic matter back to the microbial loop (“viral shunt”) and keeping blooms in check by taking down the dominant genera, such as freshwater *Microcystis* or marine *Trichodesmium*, is just a part of viral household activities (also called the “kill the winner” hypothesis) [62,63]. Field and experimental studies have shown that the fate of aquatic bacteria is roughly 50:50 due to viral lysis and protozoan bacterivory [64]. Indeed, rates of viral infection of phytoplankton by phytophages can be on par with zooplankton grazing [65]—further evidence of the virus-based bottom-up control of aquatic microbial dynamics. However, unlike protozoa and zooplankton, viruses are mostly host-specific—although some bacteriophages may have a wide range of hosts [66,67]. The net effect of the generally host-specific viral activity may be to increase or maintain biodiversity of its prey communities through its selective pressure on the most dominant, actively growing, or abundant species. Indeed, viral infections may keep microbial plankton biodiversity high while keeping overall cell densities low (below carrying capacity) in natural waters as the virus alternates between lysogenic and lytic stages [68]. In the meanwhile, the host–virus arms race continues.

Viral lysis impacts daily primary and secondary production processes by shuttling organic matter into the microbial loop via the “viral shunt”. Through means of this “viral shunt”, dissolved organic carbon is utilized in bacterial production (secondary production), is stored over the long-term as refractory dissolved organic carbon (microbial carbon pump), or recycled into CO_2_ and other inorganic nutrients (making it again available to phytoplankton for primary production)—potentially switching an ecosystem from a carbon sink into a carbon source or vice versa [64,69]. Further, freshwater mesocosm studies have shown that increasing water temperatures results in early onset of virus population dynamics—potentially altering the timing of host–virus interactions and consequently the timing of carbon and nutrient cycling [70]. This is a lot of evolutionary and biogeochemical power in some pretty small hands. When viruses flip the channel, the outcome reverberates, first within the microbial community and eventually throughout the biosphere.

## 8. One Household under Multiple Stressors

For billions of years, microorganisms have been tirelessly providing vital services to our common planetary home by making and keeping the biosphere habitable. Today, tiny microbes in the aquatic household play a disproportionately large role in the global carbon cycle. It is a role that is likely to become increasingly critical as the world’s natural waters, already under anthropogenic stress, warm up due to climate change. Humans are changing the waterscape in which the microbial housekeepers learned their trade, creating a more hostile work environment in the process. The aquatic household and the microbes therein are coming under pressure to perform their daily housekeeping tasks as a result of intensifying anthropogenic stressors (such as pollution, eutrophication, deoxygenation, and declining biodiversity) as well as climate change impacts (such as warming, acidification, and ozone depletion) [71,72].

Satellite-based studies of climate warming trends on marine productivity suggest that the tropics and mid-latitude oceans versus high-latitude oceans will respond differently to climate change. Because tropical and mid-latitude oceans are naturally nutrient-limited, increased warming leads to increased stratification, reduced mixing, even lower nutrient supply, and thus decreased plankton biomass near the sunlit surface layer [26,73]. In contrast, these authors found that high-latitude seas, where productivity is commonly light-limited due to mixing, experience somewhat enhanced productivity due to increased stratification and reduced deep mixing that retains more phytoplankton in the sunlit surface layer. Thus, different rooms of the global household may respond differently to climate change. Temporal and biogeographic changes in phytoplankton responses will in turn become reflected in downstream responses of the viral shunt, microbial loop, and classic food web with consequences on the cycling of aquatic carbon and other associated elements (Figure 6) [71,74,75]. The collective response of microbes to ongoing global warming and attendant changes might switch the hydrosphere from a net sink for atmospheric CO_2_ at present to a source of atmospheric CO_2_ in the future.

## 9. Interactive Effects of Multiple Stressors

A multitude of stressors—such as increasing CO_2_, warming, pollution, droughts, and floods—are perturbing ecosystems today [76]. Multiple stressors may act interactively on microbes with synergistic and/or antagonistic outcomes more so than simple additive effects: i.e., 1 + 2 ≠ 3 [77,78,79]. The predominance of such non-additive interactive effects of stressors poses serious challenges for predicting the response of microbes to future changes. Will a future warmer ocean become more or less productive? Some earlier experimental mesocosm studies that simulated IPCC-projected temperature increases by the end of the 21st century have found evidence that warming results in a greater increase in respiration (R) relative to production (P) and, subsequently, a decline of the P/R ratio [80]. If the house gets warmer, our microbial partners may start to eat a lot more than they produce.

Warming-driven stronger water-column stratification in the future is expected to affect the biotic structure (e.g., plankton community composition, and size-distribution) and ecosystem functioning (e.g., primary productivity and algae-bacteria relationship) because of reduced flux of nutrients from deep waters and the increased exposure of plankton to harmful Ultraviolet Radiation (UVR) in the sunlit mixed layer. In aquatic households, the tenants may be used to the stairs being closed for a while, but what if those stairs are closed for longer periods of time or permanently? Some consequences include the alteration of global phytoplankton diversity patterns, food webs, and biogeochemistry [75,81,82,83]. Further, vulnerability of microplankton to stratification also depends on other ecosystem drivers. For example, the harmful effect of UVR due to increased stratification on phytoplankton, bacteria, and their commensalistic relationship was higher in UVR-opaque than in UVR-transparent lakes. This implies that microbes of ecosystems that are opaquer to UVR may be especially vulnerable to increased thermal stratification in the future [84].

Anthropogenic addition of macro and trace nutrients, increasing CO_2_, and increasing storm frequency may have a fertilization effect on productivity [85]. However, the fertilizing effect also depends on the interaction with other factors at distinct temporal scales. At short-term (hours), shifts in primary and bacterial productivity after nutrient fertilization under UVR resulted in a reinforcement of the mutualistic control of mixotrophs on bacteria in high-mountain lakes [86]. Other studies have shown that the commensalism is enhanced or weakened depending on the interaction among nutrients, UVR, and warming [87]. At medium-term (days), heterotrophic microbes (viruses, bacteria, ciliates) and mixotrophs followed unexpected non-linear patterns of development after moderate nutrient fertilization under UVR [88,89,90]. Only at strong nutrient fertilization pulses, did obligate autotrophic algae bloom, displacing mixotrophs and heterotrophs, and, hence, lowering the taxonomic and functional biodiversity of the planktonic community [91,92]. Finally, at long-term (decades), fertilization by intensified aerosol events from desert areas under UVR and warming, also favored obligate autotrophs against mixotrophs [93]. The resulting “greening” of the formerly “blue” waters of these remote ecosystems will further alter/degrade their important ecosystem function and services.

## 10. Interrelationships among Biodiversity, Productivity, Stability, and Resiliency

Ecological studies in both terrestrial and aquatic systems have demonstrated that, in general, ecosystems display greater stability with higher levels of diversity and attendant functional redundancy [94,95,96]. Overall stability describes: (1) An ecosystem’s ability to resist a change in function or diversity following a disturbance and (2) an ecosystem’s resilience, or ability to recover, following a disturbance. The mechanism behind the diversity-stability relationship relies on the Insurance Hypothesis idea, in which an ecosystem is better equipped to maintain structure and function in the face of disturbance if a higher number of response and effect functional groups are present, ensuring functional redundancy [97,98,99]. Experiments under a multiple-stressors scenario (UVR × carbon × phosphorus interaction), have demonstrated that the metabolic balance (P/R ratio) of oligotrophic high-mountain lakes is strongly resistant to change. However, once the P/R balance underwent change, then it did not readily return back to its original state, i.e., P/R had weak resilience. Herein, the high resilience of the phototrophs—favoring their predominance over mixo- and heterotrophs—may lead to the maintenance of the net autotrophic status and carbon sink capacity of these “pristine” ecosystems [100]. Oligotrophic systems serve as sensitive bellwethers of how ecosystem stability as well as resilience are linked to their biodiversity and functional redundancy—an emergent household feature.

Within our households, we need many different members that perform a wide variety of jobs. The house is going to pile up with garbage quickly if we can recycle plastic, but not cardboard, and some are going to get quite hungry if they cannot get the food they want. Similarly, the microbial household is going to be disrupted if some vital nutrients get recycled but not others or if there is more food cooked up by the phytoplankton than there is by bacterial growth for the protozoa to eat. Diversity in the household provides a sort of ”home insurance” from major disturbances ensuring minimal damage and optimal ecosystem recovery. Indeed, extensive studies with microbial experimental systems have demonstrated that high diversity and high evenness confer functional redundancy that enhance both ecosystem stability and resiliency [96,98]. Furthermore, it is becoming apparent that microbial diversity–ecosystem function relationships are prevalent in microhabitats throughout the watershed [101]. However, with increasing intensity/frequency of climate change and anthropogenic disturbances that reduce the diversity and stability of ecosystems, they are becoming prone to regime shifts [102]. With microbes driving the biogeochemical cycles that regulate the Earth’s habitability and forming a critical component of the food web, the state and impacts of their diversity–stability–resiliency relationship deserves increased attention [103,104].

## 11. Evolving Household

The aquatic household’s physical and chemical factors exert complex selective pressures on the organisms within. Each physical variable represents a potential dimension (or trait) on which natural selection can act [105,106]. A recent global survey of phytoplankton found that temperature and environmental variability governed their diversity coincident with maximal species turnover and environmental variability [75]. The authors also came across regions of unexpectedly reduced phytoplankton diversity (relative to the otherwise generally linearly negative relationship with temperature) that were coincident with maximal species turnover and environmental variability. As an example, keeping the house at a more stable temperature helps everyone do their job better. In places where the house temperature is kept comfortable, there are a wider variety of house members, and they live there longer. Whereas in a house where the thermostat is constantly being turned to extremely hot and extremely cold, this makes most tenants frustrated in one way or another, so there is high turnover. Physical–chemical changes to the aquatic ecosystem may greatly impact how long microbes put up with the environment, who moves in and out, and how they change, adapt, survive, and evolve in their households.

Microbial communities and their diversity also represent a robust template for understanding how selective pressures can impact individual organism populations along with the potential interactions between species that are shaped in response. However, these key dynamic interactions remain uncharacterized even within controlled systems and single-species systems, such as the persistence of multiple competing strains within the long-term evolution experiment with *Escherichia coli* [107]. Such dynamics may emerge both synergistically due to sharing of ecological resources (e.g., carbon, nitrogen, and phosphorous) or antagonistically due to competition for a common limiting resource [75]. If the house thermostat is being turned up and down beyond what the microbes are used to, not only will their jobs within the household be impacted, but their interactions with other microbial members may change for better or worse. Measuring ecological dynamics and different regimes of resource allocation will be key to assessing future responses to perturbations of critical resources due to anthropogenic and climate change [71,77].

## 12. Shifting Microbiome (Communities and Their Genetic Content)

Microbial dynamics within the aquatic household represent a fascinating study system to probe questions at the interface between ecology and evolutionary genetics. Exploring and modeling these two domains together will lead to a broader understanding of evolutionary dynamics across both intra- and inter-organismic scales. Microbial ecosystems are also exciting experimental systems to perform manipulations, for instance by way of time-series studies to investigate adaptation to environmental perturbations [104,108]. When combined with emerging metagenomic sequencing technologies, this can be a powerful framework for testing a wide variety of environmental perturbations and uncovering their genetic and ecological effects [77,109]. Furthermore, in line with findings that the microbiomes (the totality of microorganisms and their genetic content) of birds across a Hawaiian valley shaped each other’s microbiomes [110], it is possible to envision such cross-microbiome interactions are likely to be similarly intense and extensive in the dynamic aquatic biome. Evidence for both latitudinal and altitudinal overlapping gradients in microbial biodiversity across the world’s watersheds are emerging [111].

Indeed, much progress is being made: The Sorcerer II Global Ocean Sampling Expedition and the Tara Oceans Expedition that circumnavigated the world have revealed millions of new genes; numerous new lineages of viruses, bacteria, archaea, microbial eukaryotes; genomes for uncultivated lineages of microbes; and provided insight into what environmental drivers shape microbial diversity and function [112,113,114]. In the future, maps of microbial households across the land–water continuum that include genomic and proteomic info should offer us insightful glimpses into both the changing structures and functions of microbial communities across the waterscape. Recent advances in Omic studies exploring the genetic fingerprints of organisms are revealing that the aquatic world is teeming with previously unknown microbial taxa representing all three of life’s major divisions (Bacteria, Archaea, and Eukarya), and a clearer picture of how microbial diversity and lifestyles impact the biosphere every day is emerging [74,115,116,117].

## 13. Sentinels of Change: Microbes as First Responders

Due to their large surface-to-volume ratios, fast reproduction time, and high reactivity, microbes are considered essential ecosystem variables that can serve as sensitive sentinels of change [51,118,119]. Additionally, natural microbial communities are the ecosystem’s domestic “first responders”, often bearing the brunt of environmental harm mitigation through their efficient breakdown of anthropogenic pollutants, leading to environmental repair and remediation. One such recent example is the key role that native microbial communities played in the breakdown of a substantial portion of the 5 million barrels of toxic oil spilled from Deepwater Horizon in 2010 [120,121,122]. However, despite their recognition as key remediators of damaged environments everywhere, the effect of anthropogenic stressors on microbes themselves is still considerably understudied [20,123].

As anthropogenic stressors and climate change impacts alter marine and freshwater ecosystems, there is increasing concern among the scientific community about the future state of the world’s natural water bodies and its repercussions on climate feedback [124,125]. Equally concerning is the human-induced loss of Earth’s biodiversity at an alarming rate—including the finding that marine animals are more vulnerable to warming than their terrestrial counterparts [126,127,128,129]. If we continue to operate with business as usual, we will inevitably alter the vast biodiversity and vital functions of the Earth’s marine and freshwater microbial housekeepers—with major spillover effects on the rest of the biosphere [72,125,126].

## 14. Unseen Strands in the Food Web: Their Housework Is Never Done

An invisible “microbial network stretches like a veil across the watery surface of our planet” whose ubiquitous presence and emergent function bear close scrutiny [1,130]. In fact, many discoveries in microbial ecology can be used to broaden public view on microorganisms and the biogeochemical cycles they drive, demonstrating their integral relationship to our personal and planet’s well-being [2,23,131,132]. It is time to reimagine the everyday microbe.

Routines of housekeeping appear to be a familiar scenario, even among life’s smallest organisms going about their everyday activities in Earth’s largest ecosystem (Figure 1 and Figure 6). Essential microbial housekeepers cook, clean and control the hydrosphere—responding actively to environmental change in globally significant ways. Thus, the housekeeping perspective is essential to the study of ecosystem dynamics at all scales.

Housekeeping activities of aquatic microorganisms slowly built the foundation of the biosphere, creating the theater in which a complex variety of life could unfold. Given that microbes have been changing Earth’s composition and in turn have been changed by Earth’s climate, they are ideal indicators of change in a world undergoing rapid environmental transformation [133]. In the future, aquatic microbes will be challenged to shoulder even greater housekeeping responsibilities in a warmer, polluted, defaunated, and deflorated household—even as their own taxonomic and physiologic diversity has taken a hit [20,75,99,134,135,136]. So, like good housekeepers everywhere, their work is never done.

## Figures and Tables

**Figure 1 life-11-00152-f001:**
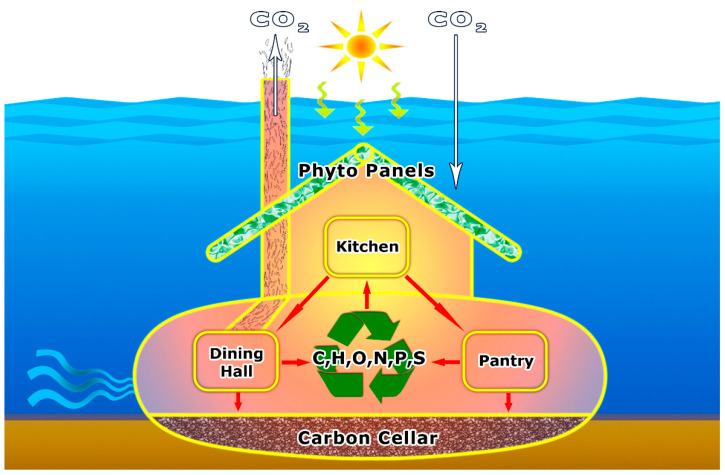
Underwater household: Simplified schematic diagram of microbial cooking, cleaning, and control in the aquatic household. Phytoplankton equipped with "Phyto Panels" prepare food in the “Kitchen” that is utilized by viruses, bacteria, archaea, protozoa, and metazoa in the “Dining Hall” or is stored in the “Pantry” for later use as dissolved and particulate organic matter. Whereas many of the elements essential to life first incorporated by phytoplankton are respired by the microbial food web or transferred to higher trophic levels, a small fraction escapes this fate and gets buried in the “Carbon Cellar” located in the sediment.

**Figure 2 life-11-00152-f002:**
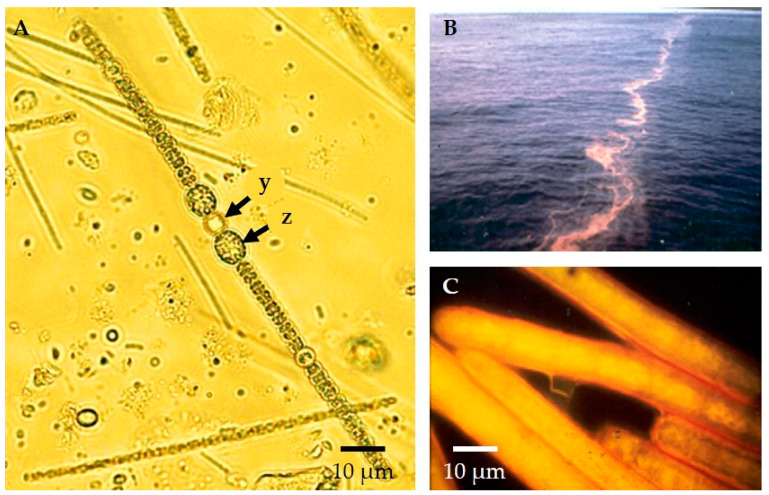
Nitrogen-fixing plankton in lakes and the ocean: (**A**) Pearl-like string of the freshwater cyanobacteria, *Dolichospermum* (formerly *Anabaena*) with a nitrogen-fixing heterocyst (y) as well as larger dormant cells called akinetes (z) as seen by brightfield microscopy. Nitrogen fixation keeps this essential nutrient biologically available, but the process is inhibited by the presence of oxygen. Specialized heterocyst cells provide an environment that is nearly oxygen-free, thus allowing *Dolichospermum* to fix atmospheric nitrogen using the enzyme, nitrogenase, while living in oxygenated waters. (**B**) Massive bloom of cyanobacterium *Trichodesmium* in the Sargasso Sea. (**C**) Nitrogen fixing bundles of marine filamentous *Trichodesmium* from the bloom under the epifluorescence microscope. Single-celled planktonic cyanobacteria ubiquitous in surface waters are also important nitrogen fixers (see below).

**Figure 3 life-11-00152-f003:**
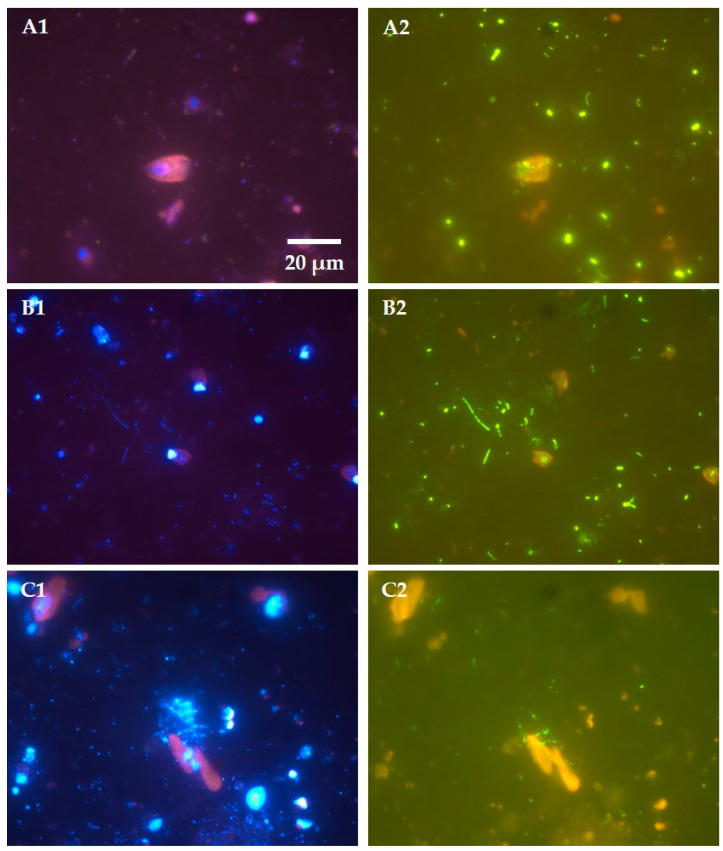
Epifluorescence microscopy images of three mixotrophic flagellates from Lake Redon (a high mountain lake in the central Pyrenees) caught in phagotrophic action. (**A**) *Cryptomonas ovata*, (**B**) *Rhodomonas minuta*, (**C**) *Dinobryon cylindricum*. Left-side images (**A1**,**B1**,**C1**) depict samples with DAPI (fluorochrome 4′,6-diamidino-2-phenylindole) staining, and right-side images (**A2**,**B2**,**C2**) depict the same samples with CARD-FISH (catalyzed reporter deposition-fluorescence in situ hybridization) staining. Images show chloroplast autofluorescence (red), DAPI-stained nucleus (blue), and free or preyed prokaryotes (blue under DAPI staining as in **A1**,**B1**,**C1**, green under CARD-FISH staining as in **A2**,**B2**,**C2**). The CARD-FISH staining was made using domain-specific probes EUB338 for *Bacteria* (**A2**,**B2**) or ARCH915 for *Archaea* (**C2**), following the procedure described in reference [38]. Note: All images are of the same magnification, and scale bar shown in image **A1** applies to all images in Figure 3.

**Figure 4 life-11-00152-f004:**
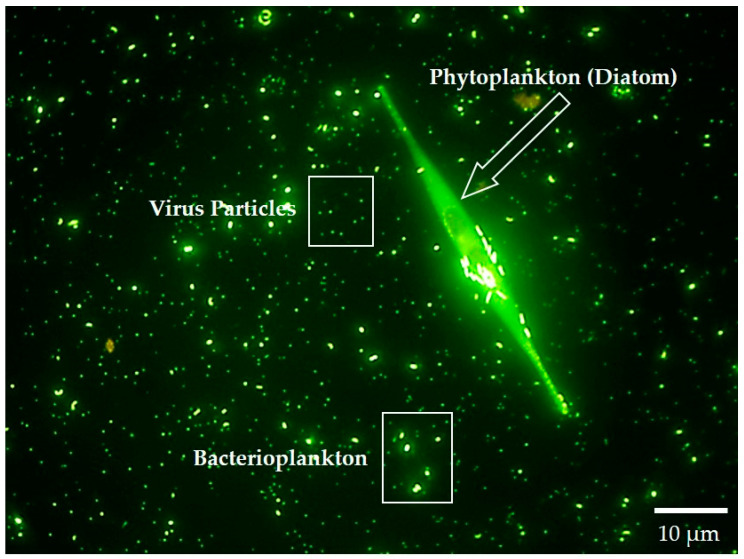
Microbial plankton of natural waters: Underwater household where bacterioplankton and a UFO-shaped diatom float in a galaxy of viruses. This image of a microbial household residing in a major Lake Michigan tributary was stained with a nucleic acid-specific fluorochrome (SYBR Green) and photographed using a fluorescence microscope at 1000× magnification. In the background are common aquatic microbes: The smallest green specs are viruses; the larger ones are comprised of heterotrophic bacteria, archaea, and photosynthetic cyanobacteria. Single-celled planktonic cyanobacteria are also important nitrogen fixers in the world’s waters. Note that photosynthetic cyanobacterial cells are usually larger than their heterotrophic bacterial companions, and some of them can be seen dividing, ~1–2 µm in diameter.

**Figure 5 life-11-00152-f005:**
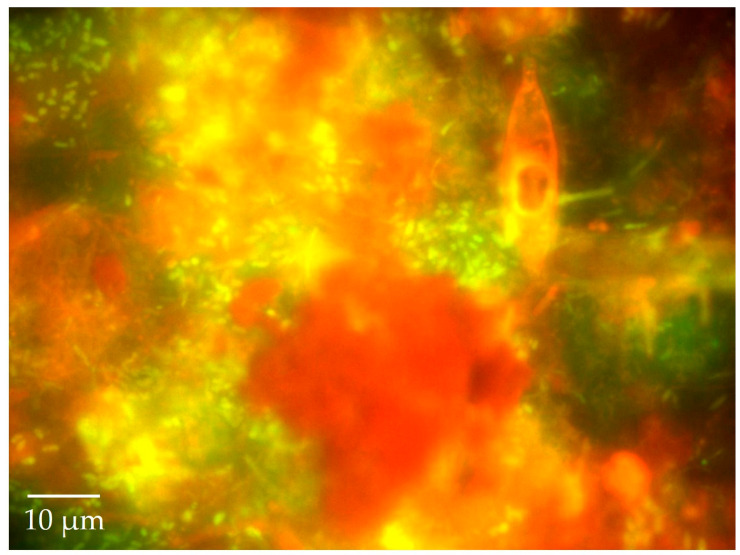
Complex underwater snow: A fluorescence image of an aggregation of bacteria, diatoms, other protists, and plant material (with chlorophyll autofluorescing red—all stuck together in a transparent exopolymer matrix commonly observed by divers as "marine snow” or “lake snow") stained with the fluorochrome Acridine Orange and observed under blue light excitation. Lake and marine snow communities are “hot spots” of intense microbial and chemical activity as they flow and fall through the water column. As a food source for some fish, lake or marine snow can directly recycle protein-rich microbial biomass to higher trophic levels. Underwater snow also serves as a vehicle for rapid transport of surface production to the sediment—a key step in the biological carbon pump.

**Figure 6 life-11-00152-f006:**
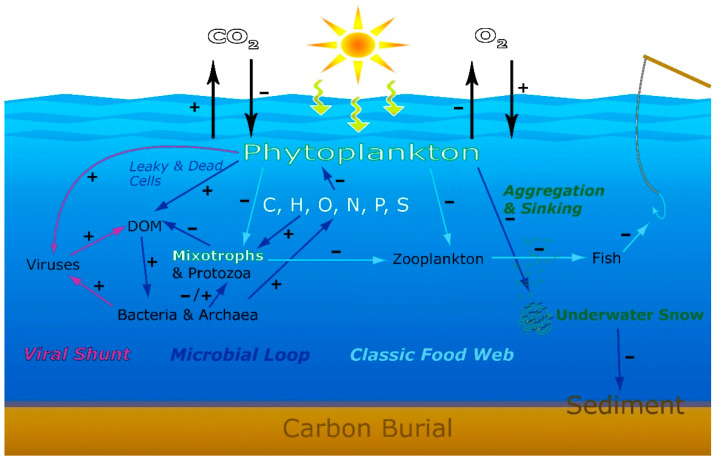
Dynamic Pathways of aquatic food webs and elemental cycles, and the expected microbial responses in a changing aquatic world. Signs + and—indicate expected increase or decrease in response to anthropogenic and climate change stressors such as changes in aquatic primary production and its role as a carbon sink. In what manner and with what intensity the many different microbial feedback processes will respond to increasing intensities of anthropogenic stressors and climate change and what net effects will result from such complex interactions remain an unresolved but consequential issue.

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
