# Peer review of "Housekeeping in the Hydrosphere: Microbial Cooking, Cleaning, and Control under Stress"

_life, 2021, doi:10.3390/life11020152_

Round 1
Reviewer 1 Report
life-1100427-peer-review-v1
Reading the title of this manuscript made me to agree providing a revision although my schedule is very tight at the moment. This housekeeping idea is a wonderful image to be compared to the fast diversity of microbially realized processes in the aquatic environment.
In the first part of the manuscript, the authors describe the roles of microbes and their realized metabolic pathways to utilize CO2, dissolved organic carbon and inorganic nutrients, the role of grazing phagotrophs and viruses and filling up the carbon cellar in the underwater household, which is all illustrated in Figure 1. In a next step, the authors take the multiple stressors that aquatic ecosystems are exposed to under a changing climate in focus and visually explain the effects and changes of microbial functions that can be expected in the future under global climate change. The vulnerability of the aquatic household (= environment) is nicely illustrated. This manuscript illustrates for a wider audience the contribution of microbial processes in aquatic environments for global biogeochemical cycles .
I very much enjoyed reading this comprehensive review about the aquatic household written in an illustrative language.
I have some minor recommendations.
Title: A large part of the manuscript deals with climate change effects on aquatic environments. Therefore, I suggest to make this issue more prominent and extent the title including the climate change idea.
L. 17, 18, 232 and elsewhere: Instead of protozoa use phagotrophic or grazing protists or microeukaryotes.
L. 104: Suggest to repeat the precise references.
Figure 2: Please indicate the method used to obtain images in a and c.
L. 163: There seems to be a typo in the sentence.
L. 164: It could be also interesting to add an image of mixotrophic protists as a major contributor to the household.
Figure 3: Please check if figure fits well in cleaning chapter. I suggest to extent the figure with some grazing protist images.
L. 298: Microcystis should be italic.
L. 312-313: Reads like a repetition of L. 297. Please rephrase.
L. 402-418: I got the impression that the housekeeping idea is missing in this paragraph.
L. 465: Suggest to use household instead of environment.
Reviewer 2 Report
Reading this review article was a real pleasure for me.
It shows the development of aquatic microbial ecology and uses a picturesque way to describe roles and functions of microbes under the ecological perspective of housekeeping. The review explains basic concepts using a special but accessible scientific language. It has a very coherent structure and synthesizes old key literature but also recent advances on the topic.
I believe that this review can speak to a large audience of researchers not so familiar to the field of aquatic microbial ecology and it can also serve as a useful educational tool.
Below you may find some minor comments:
Line 136: change to N2 (subscript)
Line 175: We could say that the same relationship exist between bacteria and heterotrophic protists i.e. bacteria use DOC released by themselves and their feeders and are eaten by them.
Line 184: mixotrophs can thrive with low bacterial prey densities but also under low light and nutrient availability
Line 229: archaeoplankton
Line 249: insert comma after competition
Line 298: Microcystis in italics
Line 385: change to CO2 (subscript)
